# A Nutraceutical Formulation Containing Brown Algae Reduces Hepatic Lipid Accumulation by Modulating Lipid Metabolism and Inflammation in Experimental Models of NAFLD and NASH

**DOI:** 10.3390/md20090572

**Published:** 2022-09-08

**Authors:** Daniela Gabbia, Marco Roverso, Ilaria Zanotto, Martina Colognesi, Katia Sayaf, Samantha Sarcognato, Diletta Arcidiacono, Alice Zaramella, Stefano Realdon, Nicola Ferri, Maria Guido, Francesco Paolo Russo, Sara Bogialli, Maria Carrara, Sara De Martin

**Affiliations:** 1Department of Pharmaceutical and Pharmacological Sciences, University of Padova, 35131 Padova, Italy; 2Department of Chemical Sciences, University of Padova, 35131 Padova, Italy; 3Department of Surgery, Oncology and Gastroenterology, University of Padova, 35131 Padova, Italy; 4Department of Pathology, Azienda ULSS2 Marca Trevigiana, 31100 Treviso, Italy; 5Gastroenterology Unit, Veneto Institute of Oncology IOV-IRCCS, 35131 Padova, Italy; 6Department of Medicine, University of Padova, 35131 Padova, Italy

**Keywords:** NAFLD, NASH, brown seaweed, algal extract, steatosis, *Ascophyllum nodosum*, *Fucus vesiculosus*

## Abstract

Recently, some preclinical and clinical studies have demonstrated the ability of brown seaweeds in reducing the risk factors for metabolic syndrome. Here, we analyzed the beneficial effect of a nutraceutical formulation containing a phytocomplex extracted from seaweeds and chromium picolinate in animal models of liver steatosis of differing severities (rats with non-alcoholic fatty liver disease (NAFLD) and its complication, non-alcoholic steatohepatitis (NASH)). This treatment led to a significant drop in hepatic fat deposition in both models (*p* < 0.01 vs. untreated animals), accompanied by a reduction in plasma inflammatory cytokines, such as interleukin 6, tumor necrosis factor α, and C reactive protein, and myeloperoxidase expression in liver tissue. Furthermore, a modulation of the molecular pathways involved in lipid metabolism and storage was demonstrated, since we observed the significant reduction of the mRNA levels of fatty acid synthase, diacylglycerol acyltransferases, the sterol-binding protein SREBP-1, and the lipid transporter perilipin-2, in both treated NAFLD and NASH rats in comparison to untreated ones. In conclusion, this nutraceutical product was effective in reducing liver steatosis and showed further beneficial effects on hepatic inflammation and glycemic control, which were particularly evident in rats characterized by a more severe condition, thus representing a therapeutic option for the treatment of NAFLD and NASH patients.

## 1. Introduction

Non-alcoholic fatty liver disease (NAFLD) is the hepatic manifestation of the metabolic syndrome (MetS) and is associated with other diseases, such as hypertension, obesity, dyslipidemia, and type 2 diabetes mellitus [1]. NAFLD is characterized by a progressive accumulation of lipids in hepatocytes in intracellular vesicles named lipid droplets. The NAFLD-associated increased risk of mortality in affected patients is mainly related to the onset of cardiovascular disorders or hepatic complications including hepatocellular carcinoma (HCC) [2].

The accumulation of fatty acids (FAs) contributes to the decline of liver function, by increasing inflammation and oxidative stress in hepatocytes and promoting their apoptosis [3]. The accumulation of free FAs (FFAs) in the liver essentially depends on the balance of three main sources, namely the hepatic uptake of non-esterified fatty acids (NEFA) from plasma, accounting for nearly 60%, de novo lipogenesis (25%) and the uptake of dietary FAs from the portal circulation (15%) [4]. In NAFLD patients, usually the excessive intake of dietary FAs, combined with an imbalance of the other two sources, is responsible for the pathological accumulation of FFAs. Moreover, other metabolic dysfunctions, e.g., obesity and insulin resistance, lead to an energy surplus saturating the storage capacity of adipose tissue, thereby increasing the release of FFAs in the bloodstream, from which they are overtaken by the liver for the synthesis of triglycerides. Furthermore, de novo lipogenesis during fasting is 3-fold higher in NAFLD patients than in healthy subjects [4], because in these patients, insulin is aberrantly stimulating this process through the sterol regulatory element binding protein (SREBP) pathway [5].

Although numerous clinical trials evaluating a variety of drug candidates are ongoing, to date there is no drug approved for the therapy of NAFLD patients. Furthermore, this disease usually has no evident clinical manifestations at the first stage, making its prompt diagnosis difficult [6]. The main clinical interventions are currently based on a nutritional approach and lifestyle changes, e.g., a calorie-restrictive diet and weight loss, aiming at the reduction of MetS risk factors and the prevention and/or reduction of NAFLD progression to severe liver failure. Since the increased prevalence of NAFLD worldwide and the detrimental consequences of the transition from NAFLD to non-alcoholic steatohepatitis (NASH) and HCC, the identification of effective therapeutic strategies to cure NAFLD and prevent its progression represents an important unmet medical need.

Several molecules/extracts of natural origin exert beneficial effects on liver function and reduce metabolic-associated risk factors. Among them, brown algae are well known due to their macro- and micronutrient content. Brown seaweeds, such as *Ascophyllum nodosum* and *Fucus vesiculosus*, are known to contain high amounts of bioactive compounds such as polysaccharides and polyphenols and are able to slow down the digestion of complex carbohydrates, improve insulin resistance, and lower the postprandial glycemic peak [7]. To further support of the efficacy of algae in the treatment and prevention of MetS-associated diseases, epidemiological evidence has shown an increased incidence of obesity and diet-induced metabolic pathologies in Western countries when compared to Eastern countries where algae are regularly consumed in many diets [7].

The intake of brown algae in healthy volunteers led to a significant reduction in plasma triglycerides in comparison to the placebo-treated group [8].

Chromium is an essential mineral involved in many metabolic pathways and biochemical responses, and can improve insulin resistance, dysregulated lipid metabolism, and liver damage [9,10]. Supplements containing chromium picolinate improve the outcome of diabetes, obesity, and polycystic ovary syndrome by regulating lipid metabolism and reducing insulin resistance and oxidative stress, all processes involved in the pathogenesis of NAFLD [11]. As a result of these peculiar effects, recent preclinical and clinical have investigated its role in NAFLD/NASH management, observing that supplementation with this mineral is useful in reducing body weight, liver steatosis, and oxidative stress in NAFLD [11,12,13].

In the light of these considerations, the aim of this study was to assess the efficacy of a commercially available nutraceutical formulation (Gdue^©^, Aesculapius Farmaceutici, Brescia, Italy), including a brown algae extract and chromium picolinate, on the hepatic lipid accumulation in two animal models with different degrees of liver steatosis obtained by the administration to Sprague Dawley male rats of a high-fat diet (HFD, 60% fat) and 30% fructose in the drinking water for 12 (NAFLD rats) or 18 (NASH rats) weeks [14].

## 2. Results

### 2.1. In Vitro Effect of the Nutraceutical Formulation on Digestive Enzymes

Since seaweeds contain molecules able to inhibit digestive enzymes devoted to the digestion of complex carbohydrates and lipids to allow their absorption in the intestinal tract, the inhibitory activity of the nutraceutical formulation on the intestinal enzymes α-lipase, α-glucosidase and α-amylase enzymes was investigated. We confirmed that the nutraceutical formulation was able to effectively inhibit both α-glucosidase and α-amylase (Figure 1A,B), as already demonstrated for the pure water extract of *F. vesiculosus* and *A. nodosum* [15], whereas no inhibitory effect was observed on the activity of α-lipase (Figure 1C), indicating that this nutraceutical formulation acts directly on carbohydrate digestion, but has probably no direct effect on the digestion of dietary lipids.

### 2.2. In Vivo Safety Assessment of the Nutraceutical Formulation on the Liver

We then performed a preliminary study in healthy male Sprague Dawley rats fed with a standard diet to evaluate the potential hepatotoxicity of a 12-week treatment with the nutraceutical formulation, which was administered once daily by oral gavage. As reported in Figure 2, no differences in the body weight increase and in liver histology at sacrifice between control and treated rats could be observed. The absence of significant differences in plasma biochemical parameters, including aspartate aminotransferase (AST), alanine aminotransferase (ALT), alkaline phosphatase (ALP), of untreated and treated rats (Table 1), confirm that the nutraceutical formulation did not cause hepatotoxic effects. Furthermore, we evaluated the concentration of three plasma markers of inflammation, i.e., interleukin-6 (IL-6), tumor necrosis factor-α (TNF-α) and C reactive protein (CRP), which were not changed by the treatment.

### 2.3. In Vivo Assessment of the Efficacy of the Nutraceutical Formulation against Liver Steatosis

#### 2.3.1. Effect on Body Weight

To assess its efficacy in reducing liver steatosis, the nutraceutical formulation was tested in vivo on rat models of NAFLD and NASH. To induce fatty liver disease of differing severities, a modified high-fat diet (HFD, 60% fat) and 30% fructose in drinking water was administered to Sprague Dawley male rats for 12 weeks to induce NAFLD, or for 18 weeks to induce NAFLD progression to NASH [16]. As reported in Figure 3 and Table 2, the treatment with the nutraceutical formulation caused a general reduction of body weight (*p* < 0.05 in both models, effect of treatment), which was increased significantly during the study (*p* < 0.0001, effect of time). A post-hoc analysis revealed that the difference between treated and untreated groups (combined effect of time and treatment) became significant after 16 weeks of administration in the NASH rats (Figure 3B).

#### 2.3.2. Effect on Physical Performance

To ascertain whether treatment had an impact on the physical performance of rats, we evaluated their coordination and resistance to fatigue by the rotarod assay, which was performed once a week during treatment. As shown in Figure 4, animals treated with the nutraceutical formulation spent on average more time on the rotarod apparatus than the untreated rats, but the difference between the two groups did not reach a level of statistical significance, probably due to high inter-individual variability. Nevertheless, this result suggests that the treatment is likely to help the improvement of the resistance to fatigue in NAFLD and NASH.

#### 2.3.3. Effect on Glycemic Control

The day before sacrifice, the animals were fed after 12 h of fasting with corn starch in sunflower oil at 50–50% to simulate a complete meal with a high content of complex carbohydrates, in combination or not with the nutraceutical formulation, given at the dose of 7.5 mg/kg. As shown in Figure 5, the treatment induced a significant decrease in postprandial blood glucose only in NASH animals. The effect on postprandial glycemia after 18 weeks of treatment is confirmed by a significant decrease in the area under the glycemic curve (AUC) and in the maximum concentration (Cmax) of blood glucose (Figure 5C). These results confirm the efficacy of the nutraceutical formulation in the reduction of dietary glucose absorption.

Since the extract of *A. nodosum* and *F. vesiculosus* improved fasting blood glucose levels in both preclinical and clinical studies [8,17,18,19], we assessed the effect of the nutraceutical formulation on fasting blood glucose, cholesterol, and triglycerides in NAFLD and NASH rats at sacrifice. As illustrated in Figure 6, the treatment significantly reduced blood glucose and plasma triglycerides in rats with NAFLD and NASH. The effect of the nutraceutical formulation on total cholesterol was significant in NAFLD rats, whereas in NASH rats this parameter was not affected by the treatment.

#### 2.3.4. Effect on Liver Function

To assess the effect of the nutraceutical formulation on liver function, some biochemical parameters, namely ALT, ALP, and total bilirubin were measured on rat plasma. As shown in Figure 7, the three biochemical markers either tended to increase or increased significantly in NAFLD and NASH rats, and the treatment with the nutraceutical formulation led to their normalization. Furthermore, the treatment reduced lipid hepatic accumulation in both NAFLD and NASH rats, as demonstrated by the quantification of triglycerides in extracts obtained from liver tissues (Figure 7D) and the histological evaluation of both the number and the size of the lipid droplets present in the liver parenchyma, stained by means of Oil Red O (ORO) (Figure 7E).

#### 2.3.5. Effect on Inflammation

Since NAFLD/NASH progression is associated with systemic inflammation [20], the plasma levels of the three pro-inflammatory markers IL-6, TNF-α, and CRP were evaluated. The results reported in Figure 8 showed an increase of the three inflammatory indexes in NAFLD rats. The treatment was able to counteract this increase, restoring their physiological levels. To evaluate the effect of the treatment on hepatic inflammation, we measured the protein expression of myeloperoxidase (MPO), a protein expressed by neutrophils, immune cells highly recruited by the liver in inflammatory conditions [21]. As expected, NAFLD and NASH rats showed a significant increase in MPO expression in liver tissue, particularly evident in rats with NASH. The treatment was able to counteract the inflammation, since MPO protein expression was significantly reduced in the hepatic parenchyma of treated rats.

#### 2.3.6. Effect on Hepatic Lipid Metabolism

Since the dysregulation of lipid metabolism and de novo lipogenesis accompanied liver steatosis, we evaluated the effect of the treatment on the expression of a panel of genes involved in these cellular pathways (Figure 9A–D). The mRNA expression of the two isoforms of diacylglycerol acyltransferase, DGAT-1 and DGAT-2, which are involved in the formation of lipid droplets, increased in NAFLD and NASH rats in comparison to the controls. The administration of the nutraceutical formulation significantly downregulated the two genes, restoring their physiological levels (Figure 9A,B). We measured the mRNA expression of the fatty acid synthase (FASN), observing a significant upregulation in NAFLD and NASH animals in comparison to the controls. The treatment positively affected FASN upregulation, significantly decreasing its mRNA expression (Figure 9C). Additionally, the sterol-binding protein SREBP-1, a transcription factor regulating lipid homeostasis, was upregulated in NAFLD rats and its physiological level was restored by the nutraceutical formulation (Figure 9D). Finally, we evaluated the protein expression of perilipin 2 (PLIN-2) in liver tissue by immunohistochemistry. PLIN-2 is involved in the storage of neutral lipids in lipid droplets and its increase has been observed in many metabolic diseases [22]. As reported in Figure 9E,F, NAFLD and NASH rats showed a significant increase in PLIN-2 protein expression in liver tissue. Interestingly, the nutraceutical treatment significantly decreased PLIN-2 protein expression in the hepatic parenchyma, especially in rats with NASH, further confirming the effect of the nutraceutical treatment on lipid metabolism and accumulation.

## 3. Discussion

NAFLD is one of the most common chronic liver diseases in Western countries, with a constantly increasing incidence worldwide [23]. Although in the early phases NAFLD is a benign and reversible condition, it could progress to more severe diseases, such as NASH, cirrhosis, and HCC. This transition occurs in consequence to a “second hit”, typically a pro-inflammatory injury of the liver [14]. The hepatic accumulation of FFAs and cholesterol induces mitochondrial dysfunction and endoplasmic reticulum (ER) stress, which leads to TNF-α-related liver damage and the increase of the production of reactive oxygen species (ROS). In the development of obesity-related NAFLD, there is a crosstalk between the liver and the adipose tissue, the latter being an endocrine and immune organ able to release in the bloodstream a variety of pro-inflammatory chemokines and adipokines, e.g., leptin, IL-6 and TNF-α, that orchestrate NAFLD development [14].

Since there is no approved pharmacological therapy for NAFLD treatment, non-pharmacological approaches, and lifestyle interventions, such as increasing physical activity and diet modifications, are currently recommended [23], including the use of nutraceutical products [7]. A previous study demonstrated that a water extract of *Ascophyllum nodosum* and *Fucus vesiculosus* improved liver function in a rat model of microvesicular steatosis [24]. In the present study, the efficacy of a commercially available nutraceutical containing a phytocomplex extracted from the same two brown algae with the addition of chromium picolinate, was evaluated on liver steatosis of differing severities, obtained by the feeding of male Sprague Dawley rats of a high-fat high-fructose diet for 12 or 18 weeks.

The nutraceutical formulation tested in this study contained a brown seaweed extract titled at 20% polyphenols and chromium picolinate (0.26 mcg/g). The presence of polysaccharides, polyphenols, and sulfolipids was confirmed by the DI-HRMS analysis of the aqueous and methanolic extract of the nutraceutical formulation (see Appendix A for further details). As already demonstrated by many preclinical and clinical studies, this study confirmed that algal extract present in this product was able to inhibit α-amylase and α-glucosidase, two key enzymes in the metabolism, thus lowering the absorption of starch and complex carbohydrates. Alpha-amylase, secreted by the pancreas and salivary glands, catalyzes the splitting of α-D-(1,4) glycosidic bonds of complex carbohydrates, e.g., starch and glycogen, lysing them into shorter oligosaccharides, whereas α-glucosidase, present in the surface membrane of the brush border of intestinal cells, activates the final phase of digestive processes by catalyzing the hydrolysis of carbohydrates and disaccharides into absorbable monosaccharides [25,26].

The inhibition of these two enzymes results in a lower assimilation of polysaccharides and a consequent reduction of postprandial blood glucose levels, as demonstrated in treated NASH animals. This inhibitory effect was not observed towards α-lipase, a gastrointestinal enzyme that plays a key role in the absorption of dietary fats by hydrolyzing triglycerides into monoglycerides and free fatty acids, which are then absorbed through the duodenal mucosa [27]. The nutraceutical formulation did not inhibit α-lipase in our experimental condition, thus suggesting that the effect of this product is mainly due to the reduction of the absorption of complex carbohydrates and not dietary fats.

The results obtained in our animal models of NAFLD, and NASH confirmed that the treatment reduces body weight gain. This decrease is likely to be related to the improvement in coordination and resistance to fatigue observed in treated rats as demonstrated by the results of rotarod test.

The results of postprandial glycemic curve also confirm the efficacy of the nutraceutical formulation in reducing glucose absorption in animals characterized by a more severe hepatic condition. Indeed, the nutraceutical treatment decreases the glycemic index in NASH animals by reducing both the AUC and Cmax of the postprandial glycemic curve, thus confirming previous data [15]. Interestingly, other data supported the effect of *F. vesicolosus* in reducing blood glucose by inhibiting the enzyme dipeptidyl peptidase-IV, which plays a key role in the degradation of incretins [28]. Moreover, the nutraceutical formulation significantly reduced the fasting plasma levels of glucose and triglycerides, which were increased in both NAFLD and NASH rats and restored the physiological levels of AST, ALP, and total bilirubin, confirming its beneficial effects on liver function. We also demonstrated a reduction of total cholesterol in rats with NAFLD. Interestingly, several preclinical studies [29] investigated the effects of brown algae on lipid levels and analyzed a possible impact on the cholesterol biosynthesis. Furthermore, it has been demonstrated that brown algae increase cholesterol excretion in feces, due to the ability of the algae compounds to bind dietary cholesterol [29]. Although the mechanisms of this cholesterol-lowering effect remain to be completely understood, carotenoids, polysaccharides, and phlorotannins, and more recently, proteins and peptides present in brown algae, have been considered to be the compounds involved in this effect. Further studies are necessary to understand the mechanism of their lowering activity on plasma cholesterol, which could be observed only in NAFLD rats in our experimental conditions.

Few studies have investigated the safety profile of algal phlorotannins; however, all of them concurred that they have low toxicity to cell cultures, invertebrates, animals, and humans [30]. In this study, the evaluation of potential hepatotoxicity performed in healthy rats treated daily with the nutraceutical formulation for 12 weeks did not evidence any sign of toxicity on hepatic tissue, thus confirming the safety of a prolonged treatment.

To evaluate the effect of this nutraceutical on hepatic steatosis, a histological analysis of liver tissue was performed by means of ORO staining, which stains the lipid droplets present in the hepatic parenchyma. An increase in the number and size of lipid droplets was observed in both NAFLD and NASH animals with respect to the control. In NASH livers, larger lipid vesicles were visible in the hepatocytes. This accumulation was effectively counteracted by the treatment. We evaluated its effect on gene expression involved in lipid metabolism and de novo lipogenesis, which are two well-known dysregulated pathways involved in NAFLD development and progression [31]. In particular, nutraceutical administration significantly downregulated DGAT1 and DGAT2 in NAFLD and NASH animals, restoring physiological levels. These two enzymes contribute to the synthesis of triacylglycerol in adipocytes and also play a crucial role in the formation of lipid droplets. In particular, DGAT-2 is located near the surface of the lipid droplets and allows the transport of triglycerides from the synthesis site to the droplets, where they accumulate and lead to their expansion [32]. Furthermore, the treatment significantly decreased the gene expression of FASN and SREBP1, which were upregulated in steatosis. The FASN enzyme is a fundamental enzyme in the endogenous lipogenesis pathway and mainly catalyzes the synthesis of long-chain fatty acids [33], whereas SREBPs regulate lipid homeostasis by controlling the transcription of many enzymes involved in the synthesis of endogenous cholesterol, fatty acids, triacylglycerols, and phospholipids [34]. To summarize, our results indicate that in steatosis the hepatic expression of DGAT-1, DGAT-2, FASN, and SREBP-1 increased significantly, and this increase was counteracted by the treatment with the nutraceutical formulation. A similar effect was observed on the protein adipophylline, also named PLIN-2, which represents one of the five proteins of the perilipin family and is related to the differentiation of adipocytes. These proteins are expressed in hepatocytes and are associated with the formation, stabilization, and degradation of lipid droplets [35,36]. PLIN-2 is upregulated in NAFLD animals and patients and promotes the accumulation of triglycerides, inhibits the oxidation of FFAs, and alters glucose tolerance [35]. PLIN-2 is upregulated in NAFLD and NASH, and the nutraceutical formulation was able to decrease PLIN-2 protein expression, particularly in the NASH model.

Another process involved in steatosis progression to NASH and more severe liver disease is the increase of oxidative stress and inflammation [37]. Thus, it has been demonstrated that the increase in ectopic fat and visceral adipose tissue promotes the secretion of some pro-inflammatory markers, including IL-6, TNF-α and PCR [38]. A study by Wang et al. demonstrated that chromium reduced the content of pro-inflammatory cytokines, such as IL-1β and TNF-α, in a murine model of hepatic steatosis [39]. Many other studies have also demonstrated that the supplementation with this essential mineral is useful in reducing inflammation and hepatic damage, and exerting hypoglycemic and hypolipidemic activities in NAFLD models and patients [40,41]. Moreover, many studies have demonstrated the anti-inflammatory properties of algal phlorotannins contained also in this algal phytocomplex [42,43,44,45], as well as their antioxidant activity [46,47,48]. The *A. nodosum* extract displayed an anti-inflammatory effect on TNF-α-challenged Caco-2 cells, by downregulating the pro-inflammatory chemokines TNF-α, IL-1β, IL-8, and IL-18 [49].

To analyze the effect of our nutraceutical formulation on the NAFLD-associated inflammation of the liver tissue, a small panel of pro-inflammatory cytokines was measured in rat plasma. IL-6 is a pro-inflammatory cytokine, whose release in the bloodstream has been related to obesity and the development of type 2 diabetes. For this reason, IL-6 has been causally related to metabolic disease [50]. TNF-α is a key mediator of systemic inflammation and is overexpressed in obesity and is considered a mediator of insulin resistance [51]. Furthermore, a causal role that greatly contributes to the development of NASH has been postulated for this cytokine [52]. CRP is a protein associated with an acute inflammatory phase and is mainly produced by the liver. Its transcriptional induction occurs in hepatocytes in response to the increased serum levels of other inflammatory cytokines, including IL-6 and TNF-α [53]. We showed that the treatment reduces the plasma levels of these inflammatory markers, which were significantly increased in NAFLD and NASH animals. Beside systemic inflammation, local inflammation in the liver has been associated with the NAFLD–NASH transition. MPO is an enzyme that belongs to the peroxidase family, abundantly expressed in immune cells such as neutrophils. These cells normally degranulate at the site of infection to counteract different types of microbial-induced disease. In the liver, neutrophils increase intracellular oxidative stress during hepatic damage. Although MPO is traditionally considered to be a neutrophil marker, it should be noticed that hepatic stellate cells and Kupffer cells also express MPO [21]. In NAFLD/NASH rats we observed a huge increase in MPO expression in liver tissue, which was counteracted by the treatment, suggesting its activity in reducing inflammatory pathways and neutrophil recruitment. In summary, the nutraceutical formulation is likely to reduce hepatic inflammation and normalize lipid metabolism and storage in both the NAFLD and NASH model.

The obtained results demonstrated that the alterations in the synthesis, metabolism, and transport of hepatic lipids observed in NAFLD and NASH rats, as well as the presence of hepatic inflammation, are effectively reduced by treatment, which inhibits the two digestive enzymes α-amylase and α-glucosidase, with a consequent reduction in carbohydrate digestion and absorption. In addition, treated animals showed beneficial effects in their liver, due to a reduced accumulation of lipids and inflammation, thereby inhibiting mechanisms involved in the progression of NAFLD and NASH.

## 4. Materials and Methods

### 4.1. In Vitro Assessement of Inhibition of Digestive Enzymes

The commercial nutraceutical product containing an algal extract of *A. nodosum* and *F. vesiculosus* (20% polyphenols) and chromium picolinate (0.26 mg/g algal extract) used in this study was supplied by Aesculapius. Its in vitro inhibitory activity on α-amylase and α-glucosidase was assessed as previously described [15]. The formulation was evaluated at increasing concentrations of 0.6, 1.5, 3, 6, 15 μg/μL for α-amylase and of 0.6, 0.3, 0.15, 0.10, 0.06 µg/µL for α-glucosidase. The inhibition on α-lipase enzymes was assessed as follows using the p-nitrophenylbutyrate (PNPB)-based assay at increasing concentrations (1, 2.5, 5, 10, 20, 40 μg/mL). Briefly, α-lipase (Sigma Aldrich, St. Louis, MI, USA) was dissolved (0.3 mg/mL) at pH 7.2, in a buffer containing 100 mM Phosphate Buffered Saline (PBS), 150 mM sodium chloride, and 0.5% Triton-X100. A total of 25 μL of each algal concentration was incubated in a 96-well plate with 50 μL of α-lipase solution, 100 μL of PBS, and 25 μL of PNPB solution. The plate was incubated for 30 min at 37 °C and the formed p-nitrophenol, proportional to the α-lipase activity, was quantified measuring the absorbance at 400 nm with a multiplate reader (Viktor Nivo, Perkin Elmer, Milan, Italy). A blank sample without α-lipase was run for each extract concentration, whereas maximum α-lipase activity was measured adding 25 μL PBS instead of extract. The % of inhibition was calculated with the following formula:$$\%\;\mathrm{inhibition}=\frac{\left(Abs\;sample-Abs\;blank\right)}{Abs\;control}\times 100$$

### 4.2. Animal Model of NAFLD/NASH

All experimental protocols were carried out with the prior authorization of the Animal Welfare Body (OPBA) of the University of Padua and the Ministry of Health (Aut. No. 101/2020-PR of 17 February 2020) and in compliance with national legislation (Legislative Decree 26/2014) and European guidelines (Directive 2010/63/EU on the protection of animals used for scientific purposes) for the handling and use of experimental animals. Male Sprague Dawley rats (6 ± 2 weeks; Charles Rivers, Italy, *n* = 10 per group) were housed in a controlled temperature and humidity room with a 12 h light–dark cycle and fed with the standard diet or with Western diet (diet rich in fat and fructose: HFHF): high fat diet (60% Kcal from fat, 23.5% from protein, 18.4% from carbohydrates; Altromin, Lage, Germany) plus 30% fructose in the drinking water. Access to diet and water was granted ad libitum for the duration of the study, except before the postprandial glucose curve was performed. Rats were randomized into experimental groups (Figure 10) and kept four per cage in the same room by the same qualified staff. During the 12- or 18-week administration of the HFHF diet, rats were treated daily by intragastric gavage with algal extract (dose 7.5 mg/kg·bw) or vehicle.

### 4.3. Postprandial Blood Glucose Levels, Biochemical and Histological Analysis

At the end of 12- or 18-w treatment, rats were fasted for 12 h and fed with a 50–50% starch and sunflower oil solution, as previously described [15]. Blood glucose levels were measured at 30, 60, 120, and 360 min by means of a glucometer (BG Star, MDSS GmbH, Hannover, Germany). To assess the degree of steatosis, livers were collected at sacrifice, and histological analysis was performed by means of Oil Red O stain (Sigma Aldrich, Merck, Darmstadt, Germany). At sacrifice, plasma samples were collected to measure alanine aminotransferase (ALT), alkaline phosphatase (ALP), and total bilirubin, triglycerides, total cholesterol, and glycemia by standard laboratory methods. The quantification of triglycerides in hepatic tissue was performed by means of a commercially available kit (ABX Pentra Triglycerides CP, Roma, Italy), following the manufacturer’s instructions.

### 4.4. Gene Expression Analysis of Hepatic Tissues

Hepatic tissue was extracted by means of the commercial kit SV Total Isolation System (Promega Corporation, Madison, WI, USA), as previously described [54]. One-step qRT-PCR was performed using the One Step SYBR PrimeScript RT-PCR kit II (Takara, Kusatsu, Japan) by means of the CFX96 Real-Time PCR Detection System (BioRad, Hercules, CA, USA), using the following thermal program: 15 min at 50 °C and 2 min at 95 °C, then 40 cycles of 15 s at 95 °C and 60 s at 60°. Primer sequences, designed by means of Primer-BLAST (NCBI, NIH), are reported in Table 3 and β-actin was used as housekeeping gene. The 2^−ΔΔCt^ method was used to quantify relative gene expression.

### 4.5. Protein Quantification of IL-6, TNF-α, and CRP in Plasma Samples

Plasma levels of IL-6 and TNF-α were measured using the Luminex xMAP^®^ technology (Luminex Corporation, Austin, TX, USA) and the “MILLIPLEX^®^ Rat Metabolic Magnetic Bead Panel” kit (Millipore, Burlington, MA, USA). Each measure was performed in duplicate. Quantitative analyses were performed with Luminex xPONENT 3.1 Software (Luminex Corporation, Austin, TX, USA) using a five-parameter logistic curve fitting.

The quantification of CRP in plasma samples was performed by means of CRP Rat ELISA Kit (Thermo Fisher Scientific, Waltham, MA, USA). Samples were quantified by measuring the absorbance at 450 nm with the multiplate reader Viktor Nivo (Perkin Elmer). CRP amount in plasma samples was calculated by non-linear regression using a calibration curve obtained with CRP standards supplied with the kit.

### 4.6. Immunohistochemical Analisys of PLIN-2 and MPO in Hepatic Tissues

A small portion of hepatic tissue was paraffin-embedded for the immunohistochemical (IHC) analysis of PLIN-2 and myeloperoxidase proteins. Then, 5 μm slices were cut with a microtome MR 2258 (Histo-Line Laboratories, Pantigliate, Milan, Italy). Antigen retrieval was performed in sodium citrate buffer pH 6 at 63 °C. Next, tissue slices were permeabilized with 0.2% Triton X100 in PBS. A total of 5% FBS was used as saturation buffer for 30 min. The following primary antibodies were used: rabbit anti-PLIN-2 (ABclonal, Woburn, MA, USA) and rabbit anti-MPO (Abcam, Cambridge, UK). The secondary HRP-conjugated polyclonal anti-rabbit IGg (KPL, Seracare, Milford, MA, USA) was used. To counterstain, nuclei hematoxylin was used. The IHC images were acquired by means of an optical microscope (Optika, Ponteranica, Bergamo, Italy) and analyzed by ImageJ software (version 1.52t, NIH, Bethesda, MD, USA).

### 4.7. Statistical Analysis

Statistical analysis was performed using GraphPad Prism (GraphPad Software Inc.; San Diego, CA, USA) ver. 8.0. by means of Student’s *t*-test, one-way or two-way ANOVA, followed by appropriated post-hoc tests, considering a *p* value < 0.05 statistically significant. Data are reported as mean ± S.E.M.

## 5. Conclusions

This study confirmed the efficacy against liver steatosis of a nutraceutical product with an inhibitory effect on the two digestive enzymes α-amylase and α-glucosidase, containing a phytocomplex extracted from brown seaweeds, in which polysaccharides, phlorotannins and other polyphenols, and sulfolipids are present. The effect on lipid accumulation is accompanied by a modulation of metabolic and inflammatory pathways, particularly evident in the liver of NASH rats. No signs of treatment-related toxicity could be evidenced in this study.

In conclusion, the present data demonstrated that this product is a valid nutraceutical support for the control and treatment of hepatic steatosis, providing support to the design of a clinical trial aiming at confirming our findings also in NAFLD and NASH patients.

## Figures and Tables

**Figure 1 marinedrugs-20-00572-f001:**
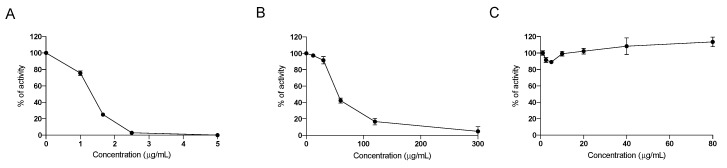
Dose-dependent inhibition of the activity of α-glucosidase (**A**), α-amylase (**B**) and α-lipase (**C**), by the nutraceutical formulation. Data are presented as means ± SEM. Results are obtained from three independent experiments run in duplicate.

**Figure 2 marinedrugs-20-00572-f002:**
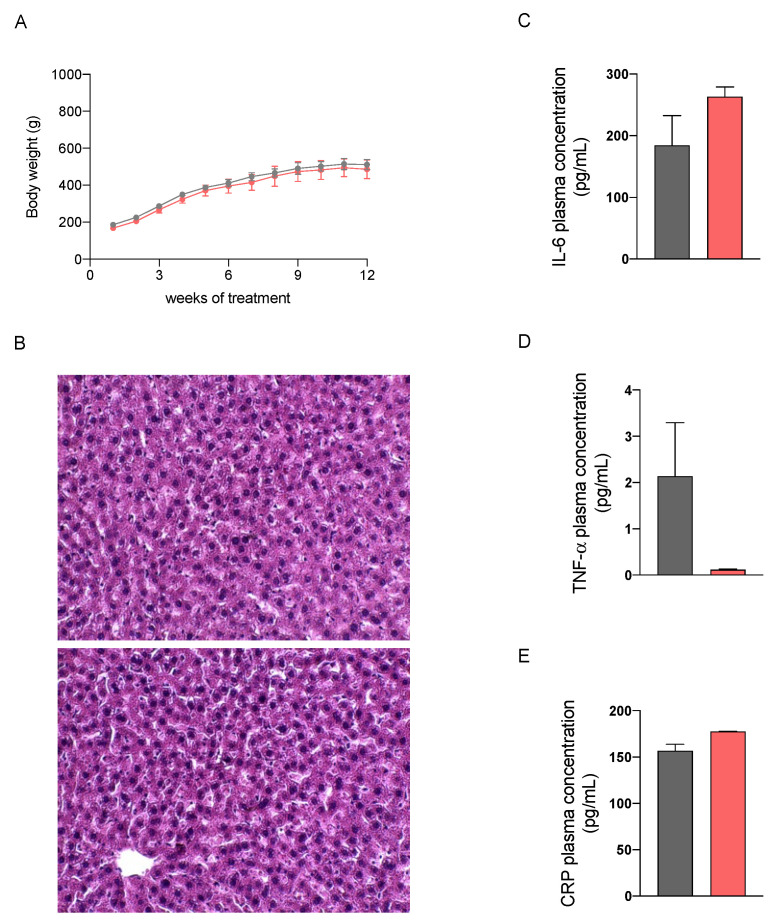
Body weight increase in healthy rats treated with vehicle (grey) or the nutraceutical formulation (light red) (**A**). Histological images of hepatic tissue, untreated (upper panel) and treated (lower panel), stained with H&E (**B**). Plasma levels of the three inflammatory markers IL6 (**C**), TNF-α (**D**) and C reactive protein (**E**). Data are presented as means ± SEM of 10 rats per group.

**Figure 3 marinedrugs-20-00572-f003:**
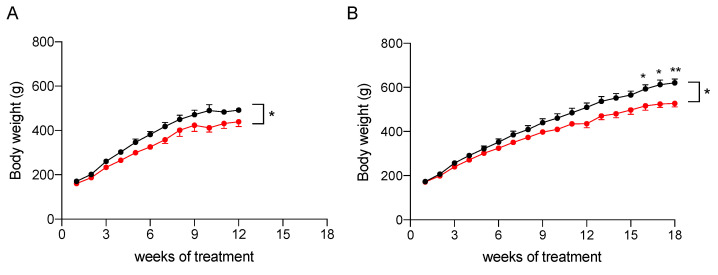
Body weight increase of NAFLD (**A**) and NASH (**B**) rats treated with vehicle (black) or the nutraceutical formulation (red). Data are presented as means ± SEM of 10 rats per group. * *p* < 0.05 and ** *p* < 0.01 vs. untreated rats.

**Figure 4 marinedrugs-20-00572-f004:**
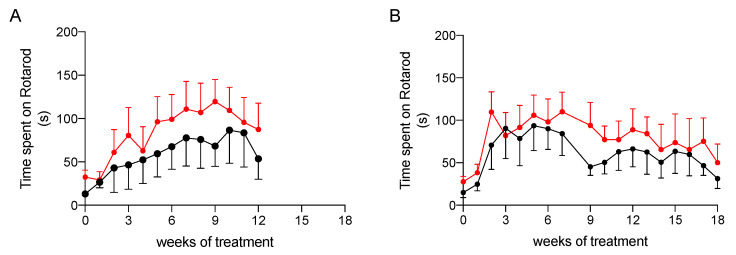
Physical performance at rotarod testing of NAFLD (**A**) and NASH (**B**) rats treated with vehicle (black) or the nutraceutical formulation (red). Data are presented as means ± SEM of 10 rats per group.

**Figure 5 marinedrugs-20-00572-f005:**
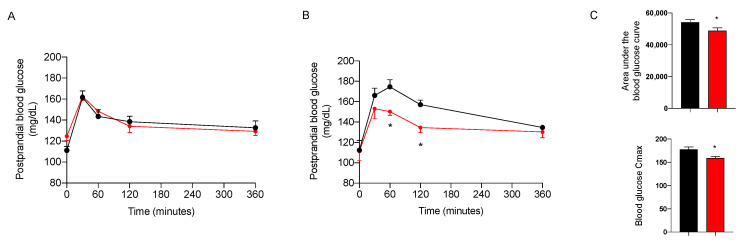
Postprandial blood glucose in NAFLD (**A**) and NASH (**B**) rats treated with vehicle (black) or the nutraceutical formulation (red). Area under the curve and Cmax of NASH rats (**C**). Data are presented as means ± SEM of 10 rats per group. * *p* < 0.05 vs. untreated NASH rats.

**Figure 6 marinedrugs-20-00572-f006:**
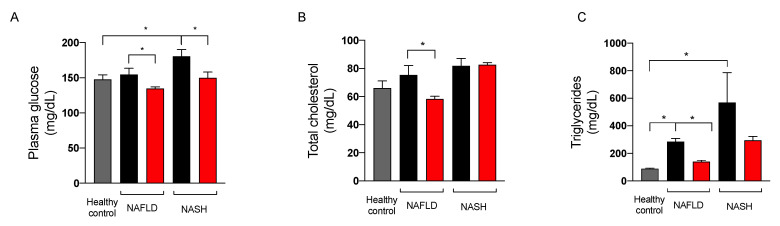
Plasma levels of glucose (**A**), total cholesterol (**B**), and triglycerides (**C**) in healthy (grey) and in NAFLD and NASH rats treated with vehicle (black) or the nutraceutical formulation (red). Data are presented as means ± SEM of 10 rats per group. * *p* < 0.05.

**Figure 7 marinedrugs-20-00572-f007:**
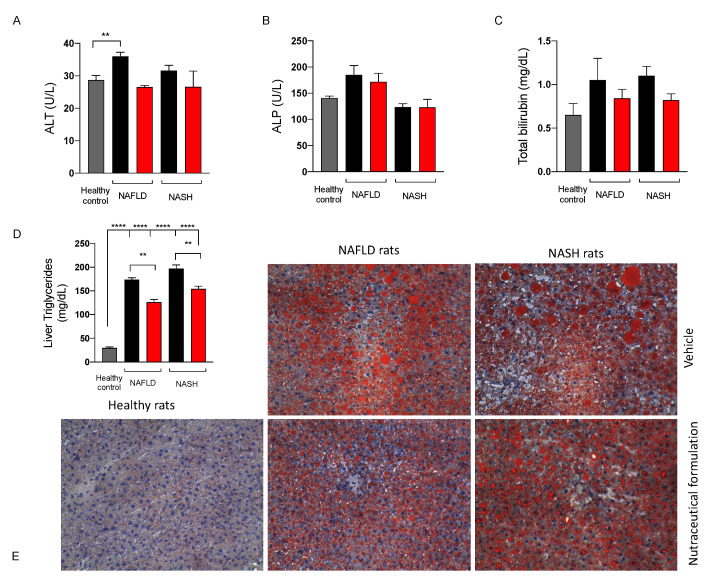
Levels of alanine aminotransferase (ALT) (**A**), alkaline phosphatase (ALP) (**B**), and total bilirubin (**C**) in plasma and triglycerides in the liver (**D**) of healthy (grey) and NAFLD and NASH treated with vehicle (black) or the nutraceutical formulation (red). Histological examination of liver steatosis by means of ORO staining (**E**), magnification 10X. Data are presented as means ± SEM of 10 rats per group. ** *p* < 0.01, **** *p* < 0.0001.

**Figure 8 marinedrugs-20-00572-f008:**
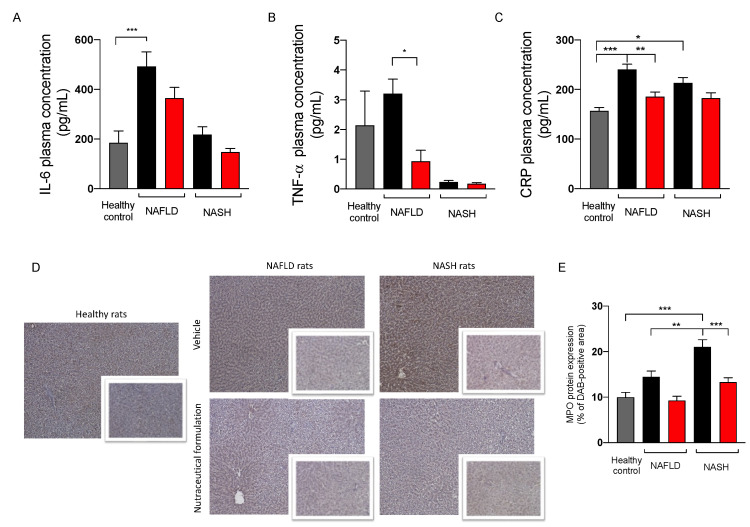
Plasma levels of IL-6 (**A**), TNF-α (**B**), and C reactive protein (**C**) in healthy (grey) and in NAFLD and NASH rats treated with vehicle (black) or the nutraceutical formulation (red). Immunohistochemical representative images of the hepatic protein expression of MPO (**D**) and their quantification (**E**). Data are presented as means ± SEM of 10 rats per group. * *p* < 0.05, ** *p* < 0.01, *** *p* < 0.001.

**Figure 9 marinedrugs-20-00572-f009:**
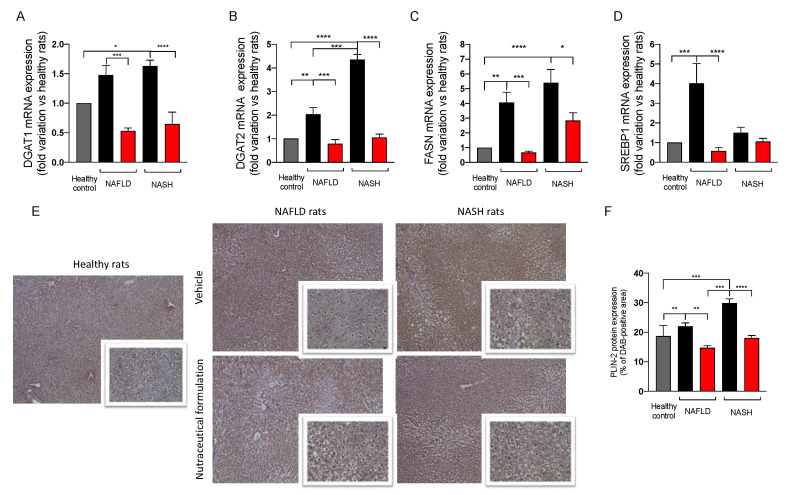
mRNA expression in liver tissue of DGAT1 (**A**), DGAT2 (**B**), FASN (**C**), SREBP1 (**D**), measured by means of qRT-PCR in healthy (grey) and in NAFLD and NASH rats treated with vehicle (black) or the nutraceutical formulation (red). Data are presented as means ± SEM of fold variation normalized on healthy rats used as controls (*n* = 10). Immunohistochemical representative images of the hepatic protein expression of PLIN-2 (**E**) and their quantification (**F**). * *p* < 0.05, ** *p* < 0.01, *** *p* < 0.001, **** *p* < 0.0001.

**Figure 10 marinedrugs-20-00572-f010:**
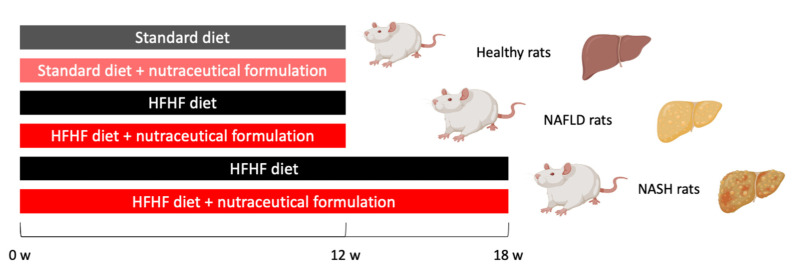
Scheme of study used for rat randomization and treatment.

**Table 1 marinedrugs-20-00572-t001:** Biochemical analysis of plasma sample collected from standard-diet-fed rats treated with vehicle or the nutraceutical formulation for 12 weeks.

Parameter	Healthy Rats (Vehicle)	Healthy Rats (Nutraceutical Formulation)
Blood glucose (mg/dL)	147.5 ± 13.1	140.3 ± 8.1
Urea (mmol/L)	5.4 ± 0.5	5.1 ± 0.2
Creatinine (μmol/L)	14.5 ± 1.9	12.7 ± 1.5
Sodium (mmol/L)	143 ± 2.2	142 ± 1.0
Potassium (mmol/L)	4.3 ± 0.5	4.2 ± 0.4
Chloride (mmol/L)	100.0 ± 2.4	100.7 ± 1.2
Total bilirubin (μmol/L)	0.7 ± 0.3	0.8 ± 0.1
Albumin (g/L)	12.0 ± 0.0	12.3 ± 0.6
Calcium (mmol/L)	2.6 ± 0.1	2.6 ± 0.1
Inorganic phosphate (mmol/L)	1.9 ± 0.2	1.8 ± 0.1
AST (U/L)	103.3 ± 17.1	94.3 ± 7.6
ALT (U/L)	29.8 ± 3.0	17.0 ± 9.5
ALP (U/L)	156.5 ± 31.4	119.0 ± 22.3
Total cholesterol (mg/dL)	66.0 ± 10.2	70.0 ± 13.5
TG (mg/dL)	89.0 ± 12.8	114.7 ± 23.6

**Table 2 marinedrugs-20-00572-t002:** Statistical analysis of body weight performed by using a Mixed Effect Model (REML).

Parameter	NAFLD Rats	NASH Rats
Time	*p* < 0.0001	*p* < 0.0001
Treatment	*p* < 0.05	*p* < 0.05
Time × Treatment	ns	*p* < 0.0001

**Table 3 marinedrugs-20-00572-t003:** Primer sequences used in the study.

Gene	Forward Primer (5′–3′)	Reverse Primer (5′–3′)
DGAT-1	TCCTGAATTGGTGCGTGGTG	GAAACAGAGACACCACCTGGA
DGAT-2	GCAGCGAGAACAAGAATAAAGGA	CCACCTTGGATCTGTTGAGC
GPAT-4	TGTGGGACGGTGGATTGAAG	GCTCCGGTCCTCATGGTTAC
FASN	GCATTTCCACAACCCCAACC	AACGAGTTGATGCCCACGAT
SREBP-1	CATGGACGAGCTACCCTTCG	GGGCATCAAATAGGCCAGGG
SREBP-2	CGAACTGGGCGATGGATGAGA	TCTCCCACTTGATTGCTGACA
β-actin	GCCACCAGTTCGCCATGGA	TTCTGACCCATACCCACCAT

## Data Availability

Raw data are available from corresponding author upon justified requests.

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
