# Peer review of "A Nutraceutical Formulation Containing Brown Algae Reduces Hepatic Lipid Accumulation by Modulating Lipid Metabolism and Inflammation in Experimental Models of NAFLD and NASH"

_marinedrugs, 2022, doi:10.3390/md20090572_

Round 1
Reviewer 1 Report
After close evaluation of the manuscript by Daniela Gabbia et al I suggest revision according to the next recommendations.
1. Please clarify abbreviations in the abstract.
2. Please provide real data of your study in the abstract.
3. The first three paragraphs of Introduction are very far away from Maribe.
4. Fucoidan from F. vesiculosus shoved remarcable DPP-IV inhivition ( https://doi.org/10.3390/md18050275) which is importany for antidiabetic potential of F. vesiculosus.
5. The rationality of doses selection is not justified, Usually, pharmacokinetic of active principles (for instance fucoidan) should be used for calculation of doses, Please address this issue,
6. Which compounds are responsible for the effects observed? This should be also mentioned in the Conclusion.
7. The reproducibility of results is one of the key aspects of science. From this point of view the object of the study should be characterized (chemical composition should be provided).
Author Response
After close evaluation of the manuscript by Daniela Gabbia et al I suggest revision according to the next recommendations.
- Please clarify abbreviations in the abstract.
We thank the reviewer for this suggestion and modified the abstract accordingly.
- Please provide real data of your study in the abstract.
We thank the reviewer for this suggestion and modified the abstract accordingly.
- The first three paragraphs of Introduction are very far away from Maribe.
We summarized the first paragraphs, leaving the information necessary for the understanding of the experiments we performed.
- Fucoidan from F. vesiculosus shoved remarcable DPP-IV inhivition ( https://doi.org/10.3390/md18050275) which is importany for antidiabetic potential of F. vesiculosus.
We thank the reviewer for this suggestion and included the citation with a brief comment in the text.
- The rationality of doses selection is not justified, Usually, pharmacokinetic of active principles (for instance fucoidan) should be used for calculation of doses, Please address this issue,
As stated in the Introduction, this nutraceutical formulation is commercially available with a recommended dosage. We based the decision of the dose by appropriate calculations to simulate the daily dose used in human patients.
- Which compounds are responsible for the effects observed? This should be also mentioned in the Conclusion.
We thank the reviewer for this suggestion and modified the conclusion accordingly.
- The reproducibility of results is one of the key aspects of science. From this point of view the object of the study should be characterized (chemical composition should be provided).
As stated in the Introduction, this nutraceutical formulation is commercially available with a recommended dosage. We used for the experiments the powder which was kindly donated by the manufacturer (Aesculapius Farmaceutici). The certificate of analysis of the algal components is attached. Furthermore, we performed an analysis of the main components which is now reported in the supplementary materials.

Reviewer 2 Report
Nonalcoholic fatty liver disease (NAFLD) is a condition in which fat builds up in liver. Nonalcoholic fatty liver (NAFL) and nonalcoholic steatohepatitis (NASH) are types of NAFLD. In addition, Nonalcoholic steatohepatitis (NASH) is liver inflammation and damage caused by a buildup of fat in the liver. This is an interesting study. There are, however, several questions need to be answered and clarified in this manuscript.
Major concern:
1. The mechanism of blood cholesterol decreasing effect after product treatment should be described in Discussion section.
2. The food intake and food efficiency should be described.
3. The abstract must be rewritten. The abstract should summarize the problem or objective of your research, and its method, results, and conclusions. It should mention each significant section of the article, with enough detail for readers to decide whether or not to read the whole paper. While it’s great to make the abstract interesting, above all it should be accurate.
4. In Figures and Tables, the animal number should be described.
5. The TG and TC levels in liver and feces should be described.
6. What kind of polyphenols rich in algal extract should be described.
7. Please draw a graphic abstract to describe how the nutraceutical formulation ameliorates the NAFLD.
Author Response
Nonalcoholic fatty liver disease (NAFLD) is a condition in which fat builds up in liver. Nonalcoholic fatty liver (NAFL) and nonalcoholic steatohepatitis (NASH) are types of NAFLD.In addition, Nonalcoholic steatohepatitis (NASH) is liver inflammation and damage caused by a buildup of fat in the liver. This is an interesting study. There are, however, several questions need to be answered and clarified in this manuscript.
Major concern:
- The mechanism of blood cholesterol decreasing effect after product treatment should be described in Discussion section.
We thank the reviewer for this important suggestion and modified the discussion accordingly.
- The food intake and food efficiency should be described.
We thank the reviewer for this comment. Unfortunately, the evaluation of the food intake was not possible, since the animals were randomized inside the cage, in order to have controls and treated animals together. This was done because the performance at the Rotarod test is often extremely influenced by the cage of origin. What we can say form our observations, is that all the cages consumed approximately the same amount of food, irrespective on the number of control and treated animals caged inside. Therefore, we do not think that the treatment has an effect on the intake of food, but this remains to be demonstrated.
- The abstract must be rewritten. The abstract should summarize the problem or objective of your research, and its method, results, and conclusions. It should mention each significant section of the article, with enough detail for readers to decide whether or not to read the whole paper. While it’s great to make the abstract interesting, above all it should be accurate.
We thank the reviewer for this suggestion and modified the abstract accordingly.
- In Figures and Tables, the animal number should be described.
The number of animals is the same in all experiments and is reported in the Method Section (n=10 rats per group). To comply with the Reviewer’s request, we added this information in the Figure legends.
- The TG and TC levels in liver and feces should be described.
We thank the Reviewer for this useful suggestion. Unfortunately, we did not collect feces, so we could not perform the analyses in this matrix. Furthermore, since it is difficult to hypothesize how cholesterol is involved in the effect we observed, and the diet we administered was not a Cholesterol-rich diet but a high-fat diet, we measured TG levels in the liver, as now reported in Fig. 6E.
- What kind of polyphenols rich in algal extract should be described.
We thank the Reviewer for this suggestion. More information about the polyphenols present in the algal extracts are now available in the supplementary materials.
- Please draw a graphic abstract to describe how the nutraceutical formulation ameliorates the NAFLD.
We thank the reviewer for this suggestion and prepared a graphical abstract.
Round 2
Reviewer 1 Report
Authors have addressed my questions and comments. I recommend publication of the manuscript in current form.
Reviewer 2 Report
I didn't find the graphic abstract in this manuscript.